
1        A Possible Pathway for Rapid Growth of Sulfate during Haze Days in China

Guohui Li[1*], Naifang Bei[2], Junji Cao[1*], Rujin Huang[1*], Jiarui Wu[1], Tian Feng[1,2], Yichen Wang[1], Suixin Liu[1],
Qiang Zhang[3], Xuexi Tie[1], and Luisa T. Molina[4]
[1]Key Lab of Aerosol Chemistry and Physics, SKLLQG, Institute of Earth Environment, Chinese Academy of
Sciences, Xi'an, China
[2]School of Human Settlements and Civil Engineering, Xi'an Jiaotong University, Xi'an, Shaanxi, China
[3]Department of Environmental Sciences and Engineering, Tsinghua University, Beijing, China
[4]Molina Center for Energy and the Environment, La Jolla, CA, and Massachusetts Institute of Technology,
Cambridge, MA, USA
[*]Correspondence to: Guohui Li (ligh@ieecas.cn), Junji Cao (jjcao@ieecas.cn), and Rujin Huang
(Rujin.Huang@psi.ch)

**Abstract**: Rapid industrialization and urbanization have caused frequent occurrence of haze
in China during wintertime in recent years. The sulfate aerosol is one of the most important
components of fine particles ($PM_{2.5}$) in the atmosphere, contributing significantly to the haze
formation. However, the heterogeneous formation mechanism of sulfate remains poorly
characterized. Observed filter measurements of $PM_{2.5}$, sulfate, and iron, and relative humidity
in Xi'an, China have been employed to evaluate the mechanism and to develop a
parameterization of the sulfate heterogeneous formation involving aerosol water for
incorporation into atmospheric chemical transport models. Model simulations with the
proposed parameterization can successfully reproduce the observed sulfate rapid growth and
diurnal variations in Xi'an and Beijing, China. Reasonable representation of sulfate
heterogeneous formation in chemical transport models considerably improves the $PM_{2.5}$
simulations, providing the underlying basis for better understanding the haze formation and
supporting the design and implementation of emission control strategies.





## 1 Introduction

Sulfate is a main component of aerosols or fine particles ($PM_{2.5}$) in the atmosphere and
plays a key role in global climate change. The direct and indirect radiative effects induced by
sulfate aerosols have constituted one of the major uncertainties in current assessments of
climate change (IPCC, 2013). In addition, deposition of sulfate aerosols exerts deleterious
impacts on ecosystems through acidification of soils, lakes, and marshes (e.g., Schindler,
1988; Gerhardsson et al., 1994). Sulfate is also an important contributor to the haze formation
and substantially reduces the atmospheric visibility during hazy days (e.g., He et al., 2014;
Guo et al., 2014).
The main source of sulfate in the atmosphere is the oxidation of sulfur dioxide ($SO_2$),
which is directly emitted from fossil fuel combustion, industrial processes, and volcanoes, or
generated by oxidation of other sulfur-containing species, such as dimethyl sulfide (DMS).
The conversion of $SO_2$ to sulfate involves various processes, including gas-phase oxidations
by hydroxyl radicals (OH) and stabilized criegee intermediates (sCI) (Mauldin III et al.,
2012), aqueous reactions in cloud or fog droplets, and heterogeneous reactions associated
with aerosols (Seinfeld and Pandis, 2006).
Model studies have been performed to investigate the formation of sulfate aerosols on
global or regional scales (Barrie et al., 2001). Previous global model results, considering the
contribution of $SO_2$ gas-phase oxidation and aqueous reactions in cloud or fog droplets driven
by ozone ($O_3$) and hydrogen peroxide ($H_2O_2$), have suggested that $SO_2$ mixing ratios are
generally overestimated while sulfate concentrations tend to be underestimated, indicating
that the two $SO_2$ oxidation pathways still cannot close the gap between field observations and
modeling studies (Kasibhatla et al., 1997; Laskin et al., 2003). Incorporation of aqueous $SO_2$
oxidation by oxygen catalyzed by transition metal ions in models has improved sulfate
simulations compared to measurements (Jacob and Hoffmann, 1983; Jacob et al., 1984, 1989;



Pandis and Seinfeld, 1992; Alexander et al., 2009), and recent studies have further shown the
enhanced role of transition metal ions catalysis during in-cloud oxidation of $SO_2$ (Harris et al.,
2013). However, models still underestimate $SO_2$ oxidation in winter source regions due to
lack of cloud or fog or a missing oxidation mechanism (Feichter et al., 1996; Kasibhatla et al.,
1997; Barrie et al., 2001). Therefore, heterogeneous conversion of $SO_2$ to sulfate associated
with aerosols provides a possible pathway for improving the sulfate simulations in chemical
transport models (CTMs) (Kasibhatla et al., 1997; Zhang et al., 2015).

Many experimental studies have been conducted to investigate the heterogeneous

reactions of $SO_2$ on various model oxides and mineral dust, but the underlying sulfate
formation mechanism is still not comprehensively understood. Generally, the complicated
sulfate heterogeneous formation from $SO_2$ is parameterized as a first-order irreversible uptake
by aerosols in CTMs, with a reactive uptake coefficient ranging from $10^{-4}$ to 0.1 and also
heavily depending on relative humidity in the atmosphere (Wang et al., 2014). It is still
imperative to develop a ubiquitous parameterization of the $SO_2$ heterogeneous reaction to
reasonably represent sulfate formation in CTMs.

In recent years, China has experienced frequently severe and persistent haze pollutions

caused by elevated $PM_{2.5}$ concentrations, and field measurements have shown that sulfate
aerosols are one of the most important species in $PM_{2.5}$ (He et al., 2014; Tian et al., 2016).
Reasonable representation of sulfate aerosols provides underlying basis for $PM_{2.5}$ simulations.
Laboratory experiments, field measurements, and model simulations have significantly
advanced our understanding of $SO_2$ heterogeneous reactions in the atmosphere, providing a
good opportunity to develop a parameterization to more reasonably represent the sulfate
formation in CTMs. In this study, a parameterization for sulfate formation from $SO_2$
heterogeneous reactions has been developed based on the daily filter measurements in Xi'an
since 2003, and verified using the Weather Research and Forecast model with Chemistry



(WRF-CHEM) in Xi'an and Beijing, China.

**2    Model and Methodology**
**2.1    WRF-CHEM Model**
In the present study, a specific version of the WRF-CHEM model (Grell et al., 2005) is
utilized to assess the proposed heterogeneous sulfate parameterization, which is developed by
Li et al. (2010, 2011a, b, 2012) at the Molina Center for Energy and the Environment. A new
flexible gas phase chemical module is incorporated into the model to consider different
chemical mechanisms, and the CMAQ/Models3 aerosol module developed by US EPA is
adopted for aerosol simulations. Chemical species surface dry depositions are parameterized
following Wesely (1989), and the wet deposition is calculated using the method in the
CMAQ. The photolysis rates are calculated using the FTUV in which the aerosol and cloud
effects on photolysis are included (Li et al., 2005; Li et al., 2011a).
The ISORROPIA Version 1.7 (Nenes et al., 1998) is used to predict inorganic aerosols
in the WRF-CHEM model. A non-traditional SOA module is employed to calculate
secondary organic aerosol (SOA) formation, including the volatility basis-set (VBS)
modeling method in which primary organic components are assumed to be semi-volatile and
photochemically reactive and are distributed in logarithmically spaced volatility bins. The
SOA contributions from glyoxal and methylglyoxal are also considered as a first-order
irreversible uptake by aerosol particles and cloud droplets in the model. Detailed information
can be found in Li et al. (2011b).
Two persistent heavy haze pollution episodes are selected in the present study: (1)
December 16 to 27, 2013 in the Guanzhong basin (GZB); and (2) January 13 to 21, 2014 in
Beijing-Tianjin-Hebei (BTH) (Figure 1). Detailed model configurations and aerosol species
observation sites are given in Table 1.





**2.2    Statistical Methods for Comparisons**

The mean bias (*MB*) and the index of agreement (*IOA*) are used to evaluate the

performance of the WRF-CHEM model in simulating gas-phase species and aerosols against
measurements. The *IOA* varies from 0 to 1, with 1 indicating perfect agreement of the
prediction with the observation.

$MB = \frac{1}{N}\sum_{i=1}^{N}(P_i - O_i)$

$IOA = 1 - \frac{\sum_{i=1}^{N}(P_i - O_i)^2}{\sum_{i=1}^{N}(|P_i - \bar{O}| + |O_i - \bar{O}|)^2}$

where $P_i$ and $O_i$ are the calculated and observed pollutant concentrations, respectively. *N* is
the total number of the predictions used for comparisons, and $\bar{P}$ and $\bar{O}$ represents the average
of the prediction and observation, respectively.
**2.3    Pollutants Measurements**

The hourly near-surface $NO_2$, $SO_2$, and $PM_{2.5}$ mass concentrations in GZB and BTH

are released by the China's Ministry of Environmental Protection (China MEP) and can be
downloaded from the website http://www.aqistudy.cn/. The daily filter measurements of
aerosol species have been performed since 2003 at the Institute of Earth Environment,
Chinese Academy of Sciences (hereafter referred to as IEECAS, 34.23°N, 108.88°E) in Xi'an,
China (Figure 1a). The sulfate, nitrate, ammonium, and organic aerosols are measured by the
Aerodyne High Resolution Time-of-Flight Aerosol Mass Spectrometer (HR-ToF-AMS) with
a novel $PM_{2.5}$ lens from 13 December 2013 to 6 January 2014 at IEECAS site in Xi'an and
from 9 to 26 January 2014 at the Institute of Remote Sensing and Digital Earth, Chinese
Academy of Sciences (40.00°N, 116.38°E) in Beijing (Figure 1b). Detailed information about
the HR-ToF-AMS measurement can be found in Elser et al. (2016).

**3    Results and Discussions**
**3.1    Parameterization of SO₂ Heterogeneous Reaction Involving Aerosol Water**



133  Figure 2 shows the scatter plot of the wintertime sulfate and $PM_{2.5}$ daily mass

134 concentrations at IEECAS from 2003 to 2010. The wintertime is defined as December of the

135 year to February of the next year. The observed daily $PM_{2.5}$ mass concentrations frequently

136 exceed 150 µg m$^{-3}$ during wintertime, showing that Xi'an has experienced heavy air pollution.

137 The sulfate aerosols constitute about 15.7% of the $PM_{2.5}$ mass concentration on average, and

138 the occurrence frequency with the daily sulfate mass concentration exceeding 50 µg m$^{-3}$ is

139 around 25.7%.

140  The observed high level of sulfate aerosols is hardly interpreted using $SO_2$ gas-phase

141 oxidations by OH and sCI due to the low $O_3$ level in the winter. The insolation is weak during

142 wintertime in North China, unfavorable for photochemical activities. The $O_3$ formation is

143 slow and the observed $O_3$ concentrations are very low, particularly during haze episodes. The

144 real-time hourly measurements of $O_3$ and $PM_{2.5}$ concentrations during 2013 and 2015

145 wintertime are analyzed as follows in GZB (5 cities, 39 sites, Figure 1a), which are released

146 by China MEP since 2013. Values of the hourly $PM_{2.5}$ concentrations ($[PM_{2.5}]$) are first

147 subdivided into 20 bins with the interval of 25 µg m$^{-3}$. $O_3$ concentrations ($[O_3]$) in the 5 cities

148 as $[PM_{2.5}]$ are assembled, and an average of $[O_3]$ in each bin are calculated (Nakajima et al.,

149 2001; Kawamoto et al., 2006). As shown in Figure 3, when $[PM_{2.5}]$ increase from 10 to 75 µg

150 m$^{-3}$, $[O_3]$ significantly decrease from around 41 to 23 µg m$^{-3}$; when $[PM_{2.5}]$ exceed 200 µg m$^{-}$

151 $^3$, $[O_3]$ fluctuate between 18 and 21 µg m$^{-3}$. The average $[O_3]$ in the 5 cities during the 2013

152 and 2015 wintertime are 27 µg m$^{-3}$ (about 13.5 ppbv). Considering the determining role of $O_3$

153 in the formation of OH and sCI in the atmosphere, the very low level of $[O_3]$ during

154 wintertime significantly reduces the efficiency of the sulfate formation from $SO_2$ oxidation

155 by OH and sCI.

156  Humid conditions have been observed to facilitate the sulfate formation in the

157 atmosphere (e.g., Sun et al., 2013; Zheng et al., 2015). Figure 4 presents the scatter plot of the



wintertime sulfate at IEECAS and the relative humidity (RH) at an adjacent meteorological
station from 2003 to 2010. The observed sulfate displays a positive correlation with the RH
with the correlation coefficient of 0.70, indicating that the aerosol water induced by the
aerosol wet growth might play an important role in the sulfate formation. It is worthy to note
that since high RH often coincides with stagnation, the concentrations of a lot of pollutants
also build up during high RH periods. There are two possible pathways for the sulfate
formation: bulk aqueous-phase oxidation of $SO_2$ in aerosol water and heterogeneous reaction
of $SO_2$ on aerosol surfaces involving aerosol water.

The heterogeneous reaction of $SO_2$ on dust surfaces has been investigated

comprehensively, but the sulfate formation mechanism is still not completely understood.
Possible mechanisms have been proposed that mineral dust and $NO_2$ enhance the conversion
of $SO_2$ to sulfate (He et al., 2014; Xie et al., 2015; Xue et al., 2016). Size-segregated particle
samples in Beijing have shown that a considerable amount of sulfate is distributed in the
coarse mode with particle diameters ranging from 2.1 to 9 μm, but sulfate peak
concentrations still occur in the fine mode with particle diameters ranging from 0.43 to 1.1
μm (Tian et al., 2016). Oxidation of sulfite by $NO_2$ in aerosol water has also been proposed to
contribute considerably to the sulfate production when $NH_3$ concentrations are sufficiently
high (Pandis and Seifeld, 1989; Xie et al., 2015).

Laboratory or field studies have suggested that $O_3$ or $Fe^{3+}$ can oxidize sulfite to sulfate.

Considering the low [$O_3$] during wintertime (Figure 3), the oxidation of sulfite by $O_3$ cannot
constitute the main source of the wintertime sulfate. Mineral dust and coal combustion in
China could provide sufficient iron. Measurements have indicated that mineral dust accounts
for about 10% of $PM_{2.5}$ in Beijing (He et al., 2014). Observations at an urban site in Ji'nan,
China have also shown enhanced iron concentrations during haze episodes, ranging from 0.7
to 5.5 μg m$^{-3}$, which are primarily emitted from steel smelting and coal combustion (Wang et





al., 2012). Figure 5 shows the scatter plot of the wintertime $PM_{2.5}$ and iron at IEECAS site
from 2003 to 2010. The iron mass concentration generally increases with $[PM_{2.5}]$, varying
from 0.1 to 10 μg m$^{-3}$, but does not correlate well with the $[PM_{2.5}]$ with the correlation
coefficient of 0.34, showing considerable background contributions. We assume that 1% of
iron in Xi'an is dissolved in aerosol water and 1% of dissolved iron is in the $Fe^{3+}$ oxidation
state (Alexander et al., 2009). When the aerosol water concentration varies from 100 to 1000
μg m$^{-3}$, the $Fe^{3+}$ concentrations in Xi'an are between 0.18 and 180 μM, providing favorable
conditions for the oxidation of adsorbed sulfite (Seinfeld and Pandis, 2006).

We propose here a $SO_2$ heterogeneous reaction parameterization in which the $SO_2$

oxidation in aerosol water by $O_2$ catalyzed by $Fe^{3+}$ is limited by mass transfer resistances in
the gas-phase and the gas-particle interface.

$$S(IV) + \frac{1}{2}O_2 \xrightarrow{Fe^{3+}} S(VI)$$

When the solution pH is between 5.0 and 7.0, the oxidation reaction is second order in
dissolved iron and first order in S(IV) and can be expressed as follows (Seinfeld and Pandis,

2006):

$$-\frac{d[S(IV)]}{dt} = 1 \times 10^{-3}[S(IV)] \quad 5.0 < pH < 6.0$$

$$-\frac{d[S(IV)]}{dt} = 1 \times 10^{-4}[S(IV)] \quad pH \sim 7.0$$

where [S(IV)] is the sulfite (S(IV)) concentration. The measured $SO_2$ mass accommodation
coefficient on aqueous surfaces is around 0.1 (Worsnop et al., 1989). Due to sufficient $NH_3$
and presence of mineral dust in the atmosphere in North China, the calculated pH in aerosol
water is between 5.0 and 7.0 (Cao et al., 2013). The $SO_2$ uptake coefficient on aerosol water
surface is estimated to be about $10^{-4} \sim 10^{-5}$ if the sulfite oxidation is catalyzed by $Fe^{3+}$. The
sulfate heterogeneous formation from $SO_2$ is therefore parameterized as a first-order
irreversible uptake by aerosols, with a reactive uptake coefficient of $0.5 \times 10^{-4}$, assuming that





there is enough alkalinity to maintain the high iron-catalyzed reaction rate:
$$\frac{d[SO_2]}{dt} = -\left(\frac{1}{4}\gamma_{SO_2}\upsilon_{SO_2}A_w\right)[SO_2]$$
where $[SO_2]$ is the $SO_2$ concentration, $A_w$ is the aerosol water surface area, $\gamma_{SO_2}$ is the $SO_2$
reactive uptake coefficient, and $\upsilon_{SO_2}$ is the $SO_2$ thermal velocity. Considering that $O_3$ and
$NO_2$ also play a considerable role in the sulfite oxidation when pH is high (Pandis and
Seinfeld, 1989), future studies are needed to consider the $O_3$ and $NO_2$ contribution to the
sulfate formation.
A box model is devised to interpret the rapid growth of sulfate observed at IEECAS
site during 2013 wintertime in Xi'an. In this model, the proposed heterogeneous reaction of
$SO_2$ involving aerosol water (hereafter referred to as HRSO$_2$) parameterization is included
and ISORROPIA (Version 1.7) is used to simulate sulfate, nitrate, ammonium aerosols, and
aerosol water. In addition, inorganic aerosols are represented by a two-moment modal
approach with a lognormal size distribution. A severe haze episode occurred from December
16 to 25, 2013 in GZB, with the average observed $[PM_{2.5}]$ exceeding 400 μg m$^{-3}$ during the
period from December 23 to 25, 2013. The HR-ToF-AMS measured sulfate concentrations
reaching about 250 μg m$^{-3}$ in the morning on December 23, and particularly, the observed
sulfate concentration increased from 132 μg m$^{-3}$ at 07:30 BJT to 240 μg m$^{-3}$ at 09:30 BJT.
The box model is utilized to simulate the rapid sulfate growth from 07:30 to 09:30 BJT,
constrained by the observed temperature, $SO_2$, nitrate, and ammonium (Table 2). There was
no RH observation at the IEECAS site; the observed RH at adjacent meteorological stations
ranged from 93% to 99% during the time period. In addition, the atmosphere was calm and
stable during the simulation period due to the control of a high pressure system over GZB, so
the horizontal transport is not considered. Various RHs from 93% to 99% are used to
calculate the sulfate growth in the box model. Figure 6 shows the calculated and observed
sulfate concentrations from 07:30 to 09:30 on December 23, 2013. The RH significantly



influences the sulfate formation and the sulfate concentrations increase nonlinearly with the
RH. When the RH is 93%, the sulfate concentration is increased by 22.7 μg m$^{-3}$ after 2-hour
integration, whereas the enhanced sulfate concentration reaches 216.6 μg m$^{-3}$ when the RH is
99%. The simulated sulfate concentrations are best fit for the observation when the RH is
98%. It is worth noting that, when RH is high (i.e., exceeding 95% or so), there is always the
possibility of the presence of fog. Studies have demonstrated that for every observed sulfate
peak in the 1980s in Los Angeles, there is fog present (Pandis and Seinfeld, 1989; Pandis et
al., 1992). Hence, the box model simulations with the RH ranging from 93% to 99% strongly
suggest that there was at least some patchy fog in the area, which would provide sufficient
water for the rapid iron-catalyzed reaction. Further studies need to be performed to
investigate the possible contributions of the patchy fog on the sulfate formation.

**3.2     Sulfate Simulations in GZB and BTH**

The proposed HRSO$_2$ parameterization is further incorporated into the WRF-CHEM

model to simulate sulfate aerosols. Two simulations are performed for GZB and BTH
respectively, including the base case (hereafter referred to as B-case) without the HRSO$_2$
parameterization and the enhanced oxidation case (hereafter referred to as E-case) with the
HRSO$_2$ parameterization. In Figures 7 and 8, we present the spatial distributions of calculated
and observed near-surface [PM$_{2.5}$] at 00:00 BJT in the E-case on selected six days
representing the haze development in GZB and BTH, respectively, along with the simulated
wind fields. In general, the predicted PM$_{2.5}$ spatial patterns agree well with the observations
at the ambient monitoring sites in GZB and BTH. The model reproduces well the high [PM$_{2.5}$]
in GZB, although it tends to underestimate the observation in the west of GZB. Due to the
specific topography, when the northeast winds are prevalent in GZB, pollutants tend to
accumulate, and simulated and observed [PM$_{2.5}$] can be up to 500 μg m$^{-3}$. When the north





winds are intensified on 26 December 2013, the pollutants commence to be transported
outside of GZB. In BTH, simulated weak winds cause severe $PM_{2.5}$ pollutions, with $[PM_{2.5}]$
frequently exceeding 250 µg m$^{-3}$ at most of areas of BTH, which is consistent with the
observations over monitoring sites. Hence, in general, the model reasonably well reproduces
the haze formation in GZB and BTH.

In the present study, ISORROPIA (Version 1.7) is employed to predict the

thermodynamic equilibrium between the sulfate-nitrate-ammonium-water aerosols and their
gas phase precursors $H_2SO_4$-$HNO_3$-$NH_3$-water vapor. $SO_2$ and $NO_2$ are the precursors of
$H_2SO_4$ and $HNO_3$, so it is imperative to evaluate the $SO_2$ and $NO_2$ simulations using the
measurements to more reasonably calculate inorganic aerosols concentrations.

Figures 9 and 10 show the temporal profiles of observed and simulated near-surface

$[NO_2]$ and $[SO_2]$ averaged over monitoring sites in GZB from December 16 to 27, 2013 and
in BTH from January 13 to 21, 2014, respectively. The model performs well in simulating the
$[NO_2]$ temporal variations compared with observations in GZB and BTH, both with the *IOA*
of 0.91 in the E-case. The difference of the simulated $[NO_2]$ in the B-case and E-case is minor,
showing that the impact of the $HRSO_2$ parameterization on $NO_2$ simulations is not significant
in GZB and BTH. Although the model replicates the temporal variations of $[SO_2]$ compared
to the measurements in GZB and NCP in the E-case with *IOAs* of around 0.80, the model
biases still exist. The model generally underestimates $[SO_2]$ in GZB and BTH, with *MBs* of -
3.4 µg m$^{-3}$ and -0.8 µg m$^{-3}$. One of the possible reasons for $SO_2$ simulation biases is that large
amounts of $SO_2$ are emitted from point sources, such as power plants or agglomerated
industrial zones, and the transport of $SO_2$ from point sources is more sensitive to the wind
field simulation uncertainties (Bei et al., 2012). The $HRSO_2$ parameterization generally
improves the $SO_2$ simulations by accelerating $SO_2$ conversions to sulfate, decreasing the *MB*
from 11.0 µg m$^{-3}$ in the B-case to -3.4 µg m$^{-3}$ in the E-case in GZB and 5.0 µg m$^{-3}$ in the B-



case to -0.8 μg m$^{-3}$ in the E-case in BTH. Overall, the model performs well in simulating the
NO$_2$ and SO$_2$ temporal variations against the measurements in GZB and BTH in the E-case.
Figures 11 and 12 display the simulated and observed inorganic aerosol variations in
Xi'an from December 16 to 27, 2013 and in Beijing from January 13 to 21, 2014,
respectively. In Xi'an, the observed sulfate mass concentrations range from 50 to 250 μg m$^{-3}$,
constituting the second most important PM$_{2.5}$ component during the episode. The HRSO$_2$
parameterization substantially improves the sulfate simulations in the E-case compared to
those in the B-case against the measurements. In the B-case, the sulfate concentrations are
remarkably underestimated, with a *MB* of -72.4 μg m$^{-3}$ (Table 3). However, in the E-case, the
WRF-CHEM model generally yields the observed sulfate variations during the 11-day
episode, with a *MB* of -17.0 μg m$^{-3}$ and an *IOA* of 0.89. It is worth noting that the model has
difficulties in reproducing the long-range transport sulfate contribution, and considerably
underestimates the observed sulfate mass concentration on December 17. The model also
cannot replicate the slow transition of synoptic situations on December 21, and the plume
formed in Xi'an is pushed to the northeast of Xi'an, causing underestimation of sulfate
aerosols (Bei et al., 2016).
In Beijing, the model also reproduces the observed sulfate variations reasonably well
during the 7-day episode in the E-case, with a *MB* of -0.8 μg m$^{-3}$ and an *IOA* of 0.88 (Table
3), but cannot adequately predict the observed sulfate peaks. The high level of sulfate
aerosols in Beijing is generally determined by the transport from surrounding areas,
particularly from the regions in the south or east. Uncertainties of the timing, depth, and
intensity of the simulated southerly or easterly wind fronts significantly influence the model
performance. For example, the early occurrence of the southerly wind fronts causes an
overestimation of sulfate aerosols during the daytime on January 15. The model also fails to
produce the observed high sulfate mass concentration in the evening during January 17 due to





the simulated weak southerly or easterly wind fronts. The improvement of sulfate simulations
caused by the $HRSO_2$ parameterization in Beijing is not as obvious as that in Xi'an due to the
very humid conditions in GZB during the simulation period, which facilitate the rapid
conversion of $SO_2$ to sulfate and cause the $SO_2$ heterogeneous conversion to be the dominant
sulfate source.

Although the *IOA* for nitrate aerosols is 0.83, the nitrate underestimation is rather large

from 17 to 21 December 2013 in Xi'an in the E-case, caused by the model failure in
simulating the long-range transport of pollutants from the east outside of GZB. The nitrate
simulations are improved in Beijing compared to those in Xi'an, with a *MB* of -4.2 μg m$^{-3}$
and an *IOA* of 0.88 in the E-case. The nitrate simulations in the B-case are slightly better than
those in the E-case, caused by the underestimation of sulfate aerosols in the B-case, which is
favorable for more $HNO_3$ to exist in the aerosol phase. The model performs well in predicting
the ammonium aerosols in Xi'an and Beijing, with *IOAs* of around 0.90 in the E-case. The
ammonium simulations in the E-case are improved compared to those in the B-case against
the measurement, showing that sulfate aerosols play a more important role in the ammonium
aerosol formation. Considering the substantial influence of simulated meteorological fields
uncertainties on the aerosol species comparison at a single site (Bei et al., 2012), the $HRSO_2$
parameterization performs reasonably well in simulating the observed inorganic aerosol
variations in Xi'an and Beijing in the E-case.

Figure 13 presents the observed and simulated diurnal cycles of mass concentrations of

$NO_2$ and $SO_2$ averaged over GZB and BTH and the sulfate, nitrate, and ammonium aerosols
in Xi'an and Beijing during the simulated episodes. The WRF-CHEM model performs well
in simulating the $NO_2$ diurnal cycles compared to measurements over GZB and BTH in the
E-case. The model also reasonably reproduces the observed diurnal cycles of $SO_2$ over GZB,
sulfate, nitrate, and ammonium aerosols in Xi'an in the E-case, particularly the sulfate





simulations are significantly improved in the E-case compared with the B-case against the
measurements. However, the model does not predict well the observed diurnal cycles of
sulfate, nitrate, and ammonium aerosols in Beijing, showing the model biases in simulating
the south or east wind fronts.

As one of the most important components of $PM_{2.5}$, reasonable representation of sulfate

heterogeneous formation in CTMs is imperative to $PM_{2.5}$ simulations and predictions. Figure
14 presents the temporal profiles of observed and simulated near-surface $[PM_{2.5}]$ averaged
over monitoring sites in GZB from December 16 to 27, 2013 and in BTH from January 13 to
21, 2014, respectively. Inclusion of the $HRSO_2$ parameterization in the E-case improves the
ability of the model to reproduce the $PM_{2.5}$ measurements in GZB and BTH. In GZB, due to
very humid conditions which facilitate the heterogeneous sulfate formation during the
episode, the simulated $PM_{2.5}$ mass concentrations are increased by more than 40 µg m$^{-3}$ in the
E-case compared to the B-case, and more consistent with the measurements. The $HRSO_2$
parameterization also improves the $PM_{2.5}$ simulations in BTH, reducing the underestimation
from around -13.3 to -5.1 µg m$^{-3}$. The $HRSO_2$ parameterization enhances considerably the
$[PM_{2.5}]$ in GZB (Figure 15), with the average $[PM_{2.5}]$ contribution of about 10 – 50 µg m$^{-3}$
from December 16 to 27, 2013. The average $[PM_{2.5}]$ contributions of the sulfate
heterogeneous formation is around 2 – 30 µg m$^{-3}$ in BTH (Figure 15) from January 13 to 21,
2014, lower than those in GZB.

The sulfate aerosol significantly affects nitrate and ammonium formation in the

atmosphere due to its stability and the deliberate thermodynamic equilibrium between
inorganic aerosols and their precursors. The simulated hourly near-surface sulfate
concentrations in E-case during the whole episode are first subdivided into 20 bins with the
interval of 5 µg m$^{-3}$. Inorganic aerosols and $PM_{2.5}$ concentrations in the B-case and E-case as
the bin sulfate concentrations in the E-case following the grid cells are assembled



respectively, and an average of inorganic aerosols and $PM_{2.5}$ concentrations in each bin are
calculated. Figures 16 and 17 show the impacts of the $HRSO_2$ parameterization on the
inorganic aerosols and $PM_{2.5}$ simulations in GZB and NCP, respectively. The heterogeneous
sulfate formation determines the sulfate level when the sulfate concentration in the E-case is
more than 25 µg m$^{-3}$, with the contribution exceeding 50% in GZB. However, in BTH, the
heterogeneous sulfate formation plays a more important role in the sulfate level only when
the sulfate concentration in the E-case exceeds 45 µg m$^{-3}$. If the $HRSO_2$ parameterization is
not considered, the model generally predicts more nitrate and less ammonium aerosols
(Figures 16b-c and 17b-c). In addition, the $[PM_{2.5}]$ contributions of the heterogeneous sulfate
formation exceed 5% and 10% when the simulated sulfate concentrations in the E-case are
more than 10 µg m$^{-3}$ and 80 µg m$^{-3}$ in GZB respectively (Figure 16d). However, in BTH, the
contributions exceed 5% when the simulated sulfate concentrations in the E-case are higher
than 50 µg m$^{-3}$ (Figure 17d).

**4    Summary and Conclusions**

In the present study, a parameterization of sulfate heterogeneous formation involving

aerosol water ($HRSO_2$) is developed based on the daily filter measurements in Xi'an since
2003. A $SO_2$ heterogeneous reaction parameterization has been proposed, in which the $SO_2$
oxidation in aerosol water by $O_2$ catalyzed by $Fe^{3+}$ is limited by mass transfer resistances in
the gas-phase and the gas-particle interface. The sulfate heterogeneous formation from $SO_2$ is
parameterized as a first-order irreversible uptake by aerosol water surfaces, with a reactive
uptake coefficient of $0.5 \times 10^{-4}$ assuming that there is enough alkalinity to maintain the high
iron-catalyzed reaction rate. A box model with the $HRSO_2$ parameterization successfully
reproduces the observed rapid sulfate formation at IEECAS site in Xi'an.

The $HRSO_2$ parameterization is implemented into the WRF-CHEM model to simulate




sulfate aerosols. Two persistent heavy haze pollution episodes are simulated with and without
the $SO_2$ heterogeneous reaction: (1) December 16 to 27, 2013 in GZB, and (2) January 13 to
21, 2014 in BTH. In general, the model performs reasonably well in simulating the $PM_{2.5}$
distributions, the $NO_2$ and $SO_2$ temporal variations compared with observations in GZB and
NCP. The $HRSO_2$ parameterization improves the $SO_2$ simulations by accelerating $SO_2$
conversions to sulfate aerosols.

The $HRSO_2$ parameterization substantially improves the sulfate simulations compared

to the measurements in Xi'an and Beijing, particularly under humid conditions. In Xi'an, the
sulfate concentrations are substantially underestimated when the $HRSO_2$ parameterization is
not considered in the simulations. Inclusion of the $HRSO_2$ parameterization significantly
enhances the sulfate formation, and the model generally produces the observed sulfate
variations during the 11-day episode. In Beijing, improvement in sulfate simulations with
$HRSO_2$ parameterization is not as obvious as that in Xi'an because of the very humid
conditions in GZB during the simulation period. The $HRSO_2$ parameterization also improves
the ammonium simulations in Xi'an and Beijing compared to observations, as well as
appreciably improves the $PM_{2.5}$ simulations against the measurements over monitoring sites
in GZB and NCP.

In summary, reasonable representation of sulfate heterogeneous formation not only

improves the $PM_{2.5}$ simulations, but also helps rationally verify the contribution of inorganic
aerosols to $PM_{2.5}$, providing the underlying basis for better understanding the haze formation
and supporting the design and implementation of emission control strategies.

Data availability: The real-time $NO_2$, $SO_2$ and $PM_{2.5}$ are accessible for the public on the
website http://106.37.208.233:20035/. The historic profile of observed ambient pollutants is
also available at http://www.aqistudy.cn/.



*Acknowledgements.* The authors would like to acknowledge helpful discussion with Professor
Spyros Pandis. This work was supported by the National Natural Science Foundation of
China (No. 41275153) and supported by the "Strategic Priority Research Program" of the
Chinese Academy of Sciences,Grant No. XDB05060500. Guohui Li is also supported by
the "Hundred Talents Program" of the Chinese Academy of Sciences. Naifang Bei is
supported by the National Natural Science Foundation of China (No. 41275101).



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



**Figure Captions**

Figure 1 WRF-CHEM simulation domains with topography for (a) the Guanzhong basin and
(b) Beijing-Tianjin-Hebei. The black squares represent ambient monitoring sites. The
red filled circles in (a) and (b) denote the deployment locations of the HR-ToF-AMS
in Xi'an and Beijing, respectively.

Figure 2 Scatter plot of the observed sulfate with $PM_{2.5}$ mass concentrations at IEECAS site
in Xi'an during the wintertime from 2003 to 2011.

Figure 3 Average $O_3$ mass concentrations over monitoring sites in GZB as a function of the
$PM_{2.5}$ mass concentration during the wintertime from 2013 to 2015.

Figure 4 Scatter plot of the observed relative humidity with sulfate mass concentrations at
IEECAS site in Xi'an during the wintertime from 2003 to 2011.

Figure 5 Scatter plot of the observed $PM_{2.5}$ with iron mass concentrations at IEECAS site in
Xi'an during the wintertime from 2003 to 2011.

Figure 6 Sulfate growth simulated by the box model with the $HRSO_2$ parameterization under
various relative humidity conditions at IEECAS site in Xi'an from 07:30 to 09:30 BJT
on December 23, 2013. The black dots denote the HR-ToF-AMS measurement and
the solid lines with different colors represent the box model simulations under
different relative humidity.

Figure 7 Pattern comparison of simulated vs. observed near-surface $PM_{2.5}$ at 00:00 BJT
during the selected six days in GZB from 16 to 27 December 2013. Colored circles:
$PM_{2.5}$ observations; color contour: $PM_{2.5}$ simulations in the E-case; black arrows:
simulated surface winds in the E-case.

Figure 8 Same as Figure 7, but in BTH from 13 to 21 January 2014.
Figure 9 Comparison of measured and predicted diurnal profiles of near-surface hourly (a)
$NO_2$ and (b) $SO_2$ averaged over all ambient monitoring sites in GZB from 16 to 27
December 2013. The black dots correspond to the observations, and the solid red and
blue lines are the simulations in the E-case and B-case, respectively.

Figure 10 Same as Figure 9, but in BTH from 13 to 21 January 2014.
Figure 11 Comparison of measured and simulated diurnal profiles of inorganic aerosols of (a)
sulfate, (b) nitrate, and (c) ammonium in Xi'an from 16 to 27 December 2013. The
black dots represent the observations, and the solid red and blue lines denote the
simulations in the E-case and B-case, respectively.

Figure 12 Same as Figure 11, but in Beijing from 13 to 21 January 2014.
Figure 13 Observed and simulated diurnal cycles of mass concentrations of $NO_2$ and $SO_2$
averaged over GZB and BTH and the sulfate, nitrate, and ammonium aerosols in
Xi'an and Beijing during the simulated episodes.

Figure 14 Comparison of measured and predicted diurnal profiles of near-surface hourly
$PM_{2.5}$ mass concentration averaged over all ambient monitoring stations (a) in GZB





from 16 to 27 December 2013 and (b) in BTH from 13 to 21 January 2014. The black
dots represent the observations, and the solid red and blue lines are the simulations in
the E-case and B-case, respectively.
Figure 15 Distribution of the average near-surface $PM_{2.5}$ contribution due to the $SO_2$
heterogeneous reactions in GZB and BTH during the simulated episodes.
Figure 16 Average (a) sulfate, (b) nitrate, (c) ammonium, and (d) $PM_{2.5}$ mass concentrations
in GZB during the simulation period as a function of the sulfate mass concentration in
the E-case. The red and blue dots represent average mass concentrations in the E-case
and B-case, respectively.
Figure 17 Same as Figure 16, but in BTH from 13 to 21 January 2014.




Table 1 WRF-CHEM model configurations and observation sites

| Regions | Guanzhong Basin (GZB) | Beijing-Tianjin-Hebei (BTH) |
|---|---|---|
| Simulation period | December 16 to 27, 2013 | January 13 to 21, 2014 |
| Domain size | 150 × 150 | |
| Domain center | 34.25°N, 109°E | 39°N, 117°E |
| Horizontal resolution | 6km × 6km | |
| Vertical resolution | 35 vertical levels with a stretched vertical grid with spacing ranging from 30 m near the surface, to 500 m at 2.5 km and 1 km above 14 km | |
| Microphysics scheme | WSM 6-class graupel scheme (Hong and Lim, 2006) | |
| Boundary layer scheme | MYJ TKE scheme (Janjić, 2002) | |
| Surface layer scheme | MYJ surface scheme (Janjić, 2002) | |
| Land-surface scheme | Unified Noah land-surface model (Chen and Dudhia, 2001) | |
| Longwave radiation scheme | Goddard longwave scheme (Chou and Suarez, 2001) | |
| Shortwave radiation scheme | Goddard shortwave scheme (Chou and Suarez, 1999) | |
| Meteorological boundary and initial conditions | NCEP 1°×1° reanalysis data | |
| Chemical initial and boundary conditions | MOZART 6-hour output (Horowitz et al., 2003) | |
| Anthropogenic emission inventory | Developed by Zhang et al. (2009) | |
| Biogenic emission inventory | MEGAN model developed by Guenther et al. (2006) | |
| **Aerosol Observation Sites** | | |
| City | Xi'an | Beijing |
| Longitude and latitude | 34.23°N, 108.88°E | 40.00°N, 116.38°E |




Table 2 Box model configurations

| Time (BJT) | 07:00 – 08:00 | 08:00 – 09:00 | 09:00 – 10:00 |
|---|---|---|---|
| Temperature ($^{\circ}$C) | -3.7 | -3.2 | -2.1 |
| SO$_2$ concentration (µg m$^{-3}$) | 10.7 | 10.4 | 25.5 |
| Nitrate concentration (µg m$^{-3}$)[*] | 67.6 | 70.1 | 69.1 |
| Ammonium concentration (µg m$^{-3}$)[*] | 65.2 | 76.0 | 91.9 |

[*]The HR-ToF-AMS measures sulfate, nitrate, and ammonium aerosols with a time resolution of 1 minute. The
high temporal resolution nitrate and ammonium are used to constrain the box model and the hourly average is
presented in the table.





Table 3 Statistical comparisons of simulated and measured sulfate, nitrate, and ammonium
concentrations in Xi'an and Beijing

| City | Species | E-case | | B-case | |
|------|---------|--------|------|--------|------|
| | | $MB$ (µg m$^{-3}$) | $IOA$ | $MB$ (µg m$^{-3}$) | $IOA$ |
| Xi'an | Sulfate | -17.0 | 0.89 | -72.4 | 0.50 |
| | Nitrate | -13.4 | 0.83 | -6.3 | 0.86 |
| | Ammonium | -5.1 | 0.92 | -20.1 | 0.72 |
| Beijing | Sulfate | -0.8 | 0.88 | -8.4 | 0.65 |
| | Nitrate | -4.2 | 0.88 | -1.9 | 0.92 |
| | Ammonium | -2.7 | 0.89 | -4.1 | 0.87 |






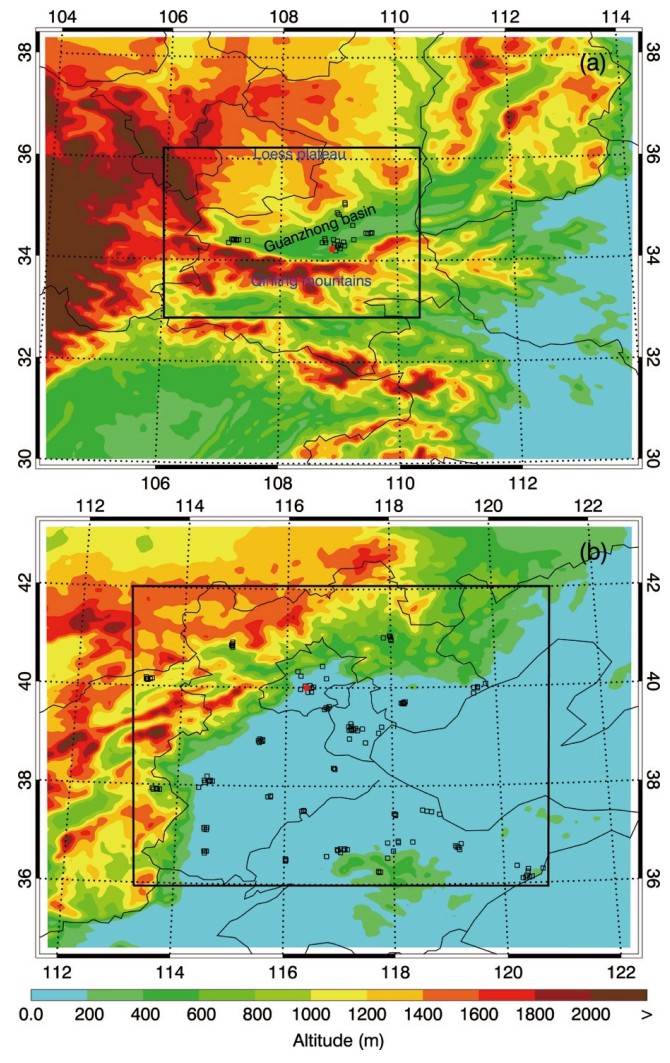

Figure 1 WRF-CHEM simulation domains with topography for (a) the Guanzhong basin and
(b) Beijing-Tianjin-Hebei. The black squares represent ambient monitoring sites. The red
filled circles in (a) and (b) denote the deployment locations of the HR-ToF-AMS in Xi'an
and Beijing, respectively.





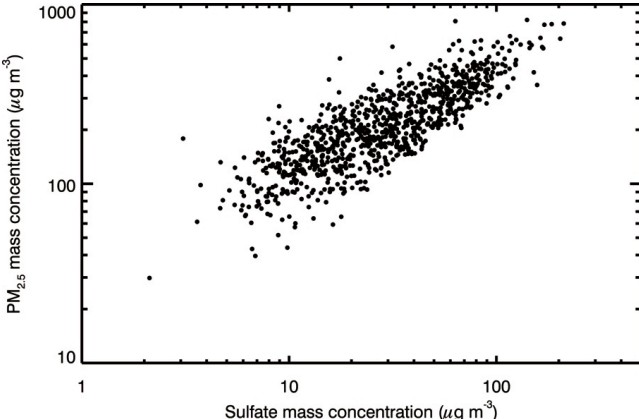

Figure 2 Scatter plot of the observed sulfate with PM$_{2.5}$ mass concentrations at IEECAS site
in Xi'an during the wintertime from 2003 to 2011.



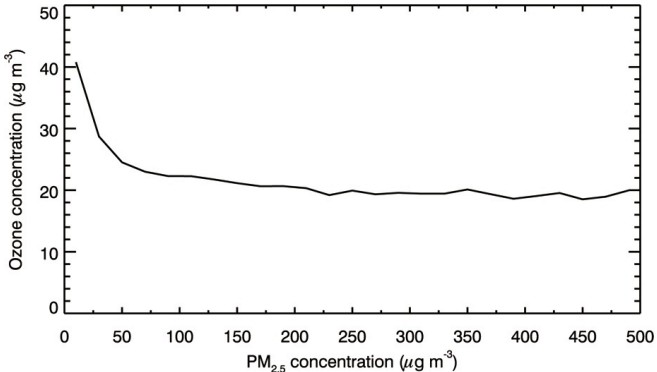



Figure 3 Average $O_3$ mass concentrations over monitoring sites in GZB as a function of the
$PM_{2.5}$ mass concentration during the wintertime from 2013 to 2015.








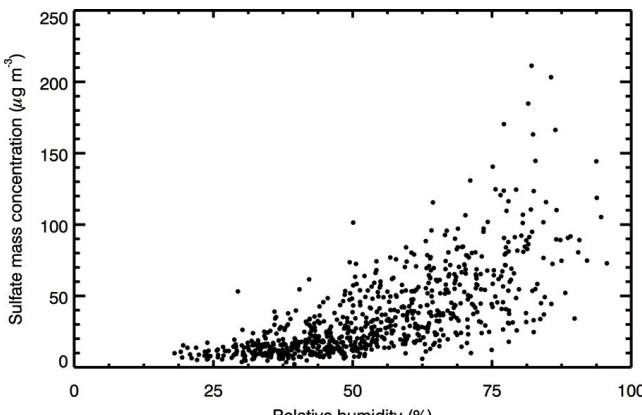

Figure 4 Scatter plot of the observed relative humidity with sulfate mass concentrations at
IEECAS site in Xi'an during the wintertime from 2003 to 2011.





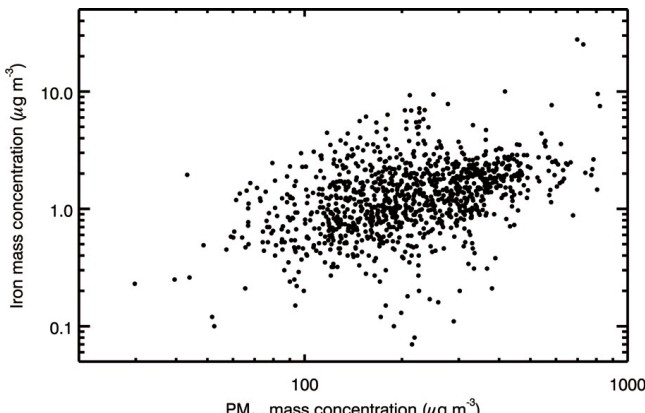

Figure 5 Scatter plot of the observed PM$_{2.5}$ with iron mass concentrations at IEECAS site in
Xi'an during the wintertime from 2003 to 2011.





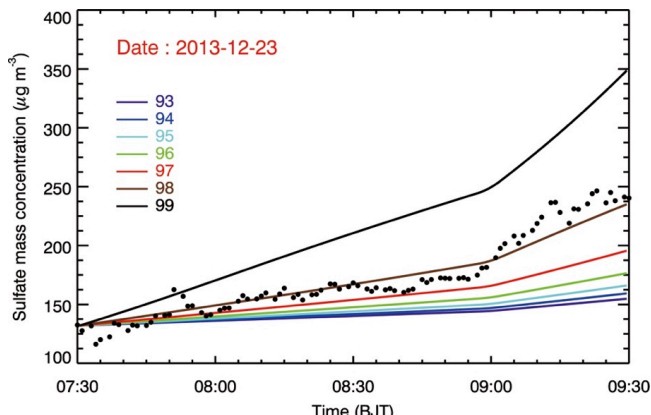

Figure 6 Sulfate growth simulated by the box model with the $HRSO_2$ parameterization under
various relative humidity at IEECAS site in Xi'an from 07:30 to 09:30 BJT on December 23,
2013. The black dots denote the HR-ToF-AMS measurement and the solid lines with
different colors represent the box model simulations under different relative humidity.







Figure 7 Pattern comparison of simulated vs. observed near-surface $PM_{2.5}$ at 00:00 BJT on
the selected six days in GZB from 16 to 27 December 2013. Colored circles: $PM_{2.5}$
observations; color contour: $PM_{2.5}$ simulations in the E-case; black arrows: simulated surface
winds in the E-case.





Figure 8 Same as Figure 7, but in BTH from 13 to 21 January 2014.





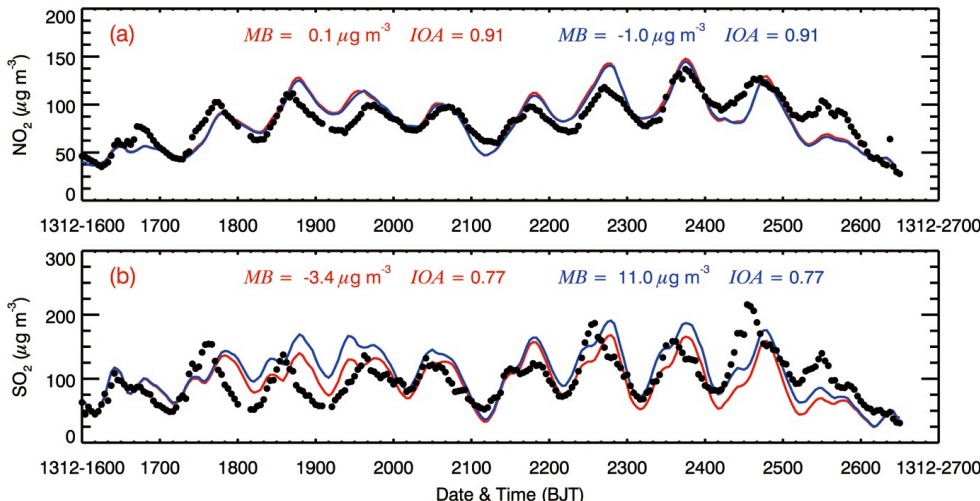

Figure 9 Comparison of measured and predicted diurnal profiles of near-surface hourly (a) $NO_2$ and (b) $SO_2$ averaged over all ambient monitoring sites in GZB from 16 to 27 December 2013. The black dots correspond to the observations, and the solid red and blue lines are the simulations in the E-case and B-case, respectively. The x-axis labels (named date and time) represent year, month, day and hour (YYMM-DDHH) or day and hour (DDHH). For example, 1312-1600 represents 00 BJT on 16 December 2013.



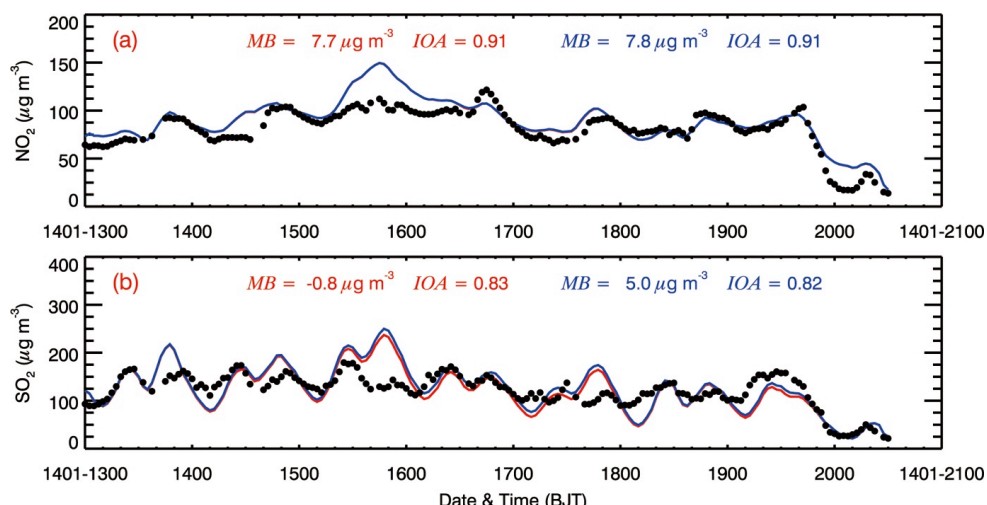

Figure 10 Same as Figure 9, but in BTH from 13 to 21 January 2014.





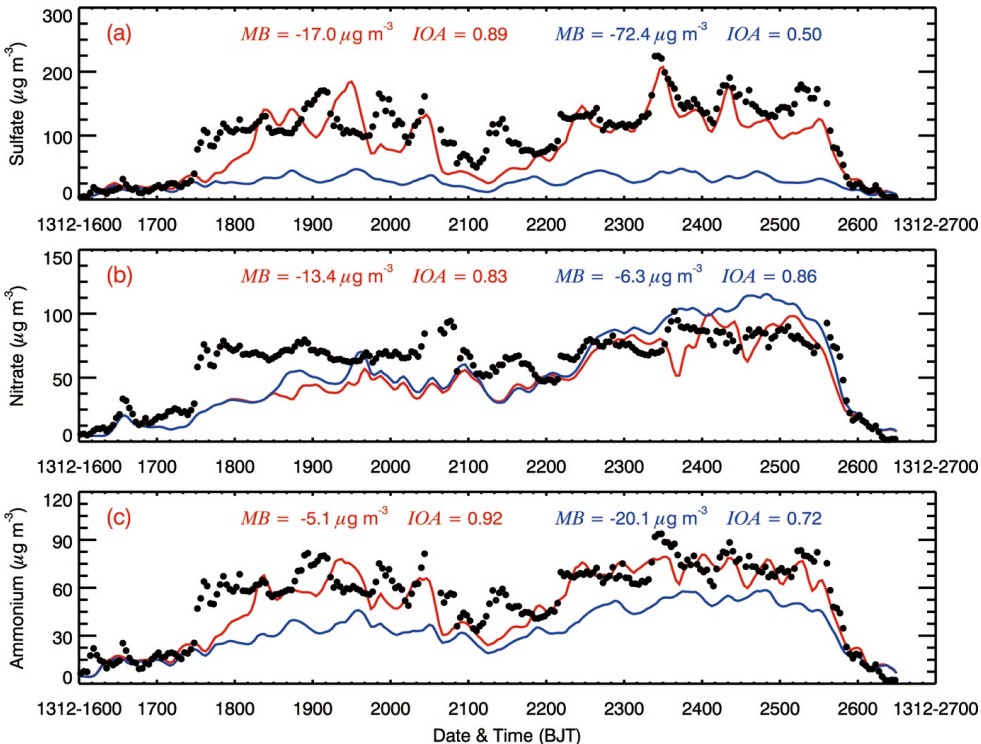

Figure 11 Comparison of measured and simulated diurnal profiles of inorganic aerosols of (a) sulfate, (b) nitrate, and (c) ammonium in Xi'an from 16 to 27 December 2013. The black dots represent the observations, and the solid red and blue lines denote the simulations in the E-case and B-case, respectively.



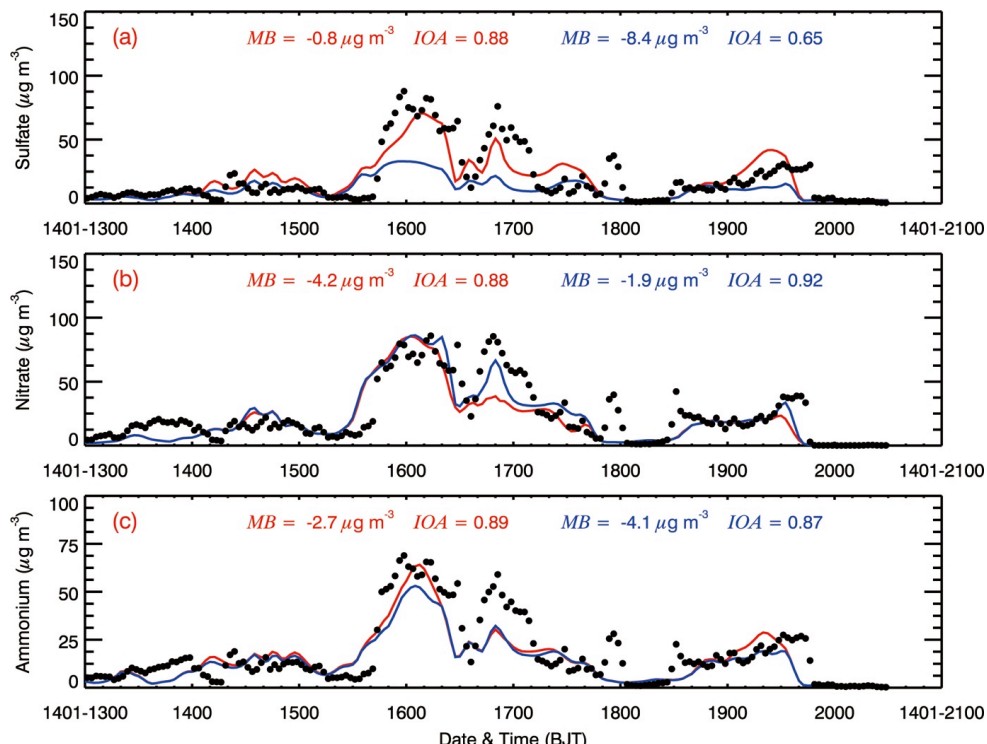

Figure 12 Same as Figure 11, but in Beijing from 13 to 21 January 2014.



Figure 13 Observed and simulated diurnal cycles of mass concentrations of $NO_2$ and $SO_2$
averaged over GZB and BTH and the sulfate, nitrate, and ammonium aerosols in Xi'an and
Beijing during the simulated episodes.




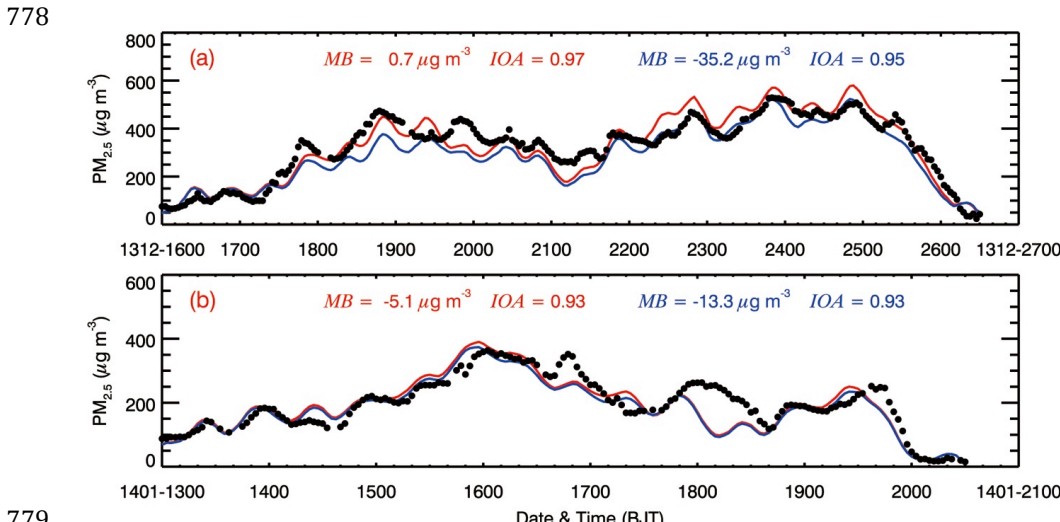

Figure 14 Comparison of measured and predicted diurnal profiles of near-surface hourly
PM$_{2.5}$ mass concentration averaged over all ambient monitoring stations (a) in GZB from 16
to 27 December 2013 and (b) in BTH from 13 to 21 January 2014. The black dots represent
the observations, and the solid red and blue lines are the simulations in the E-case and B-case,
respectively.





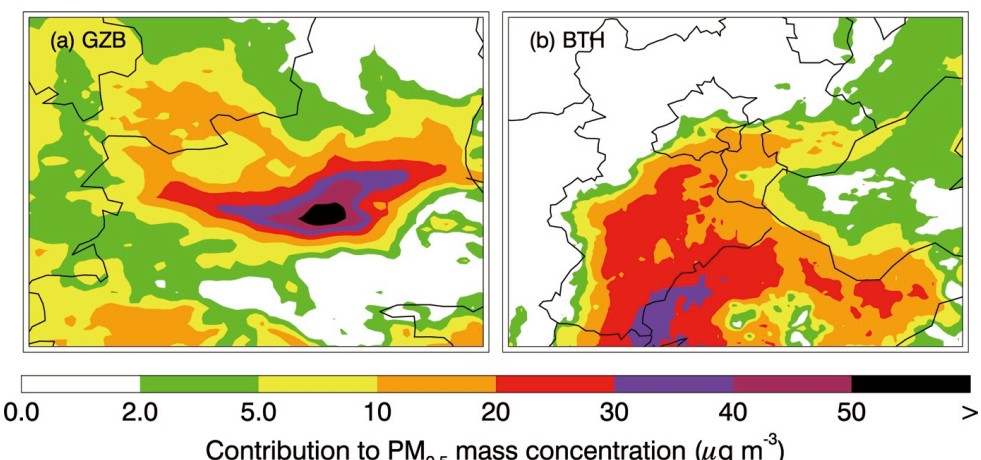

Figure 15 Distribution of the average near-surface $PM_{2.5}$ contribution due to the $SO_2$
heterogeneous reactions in GZB and BTH during the simulated episodes.


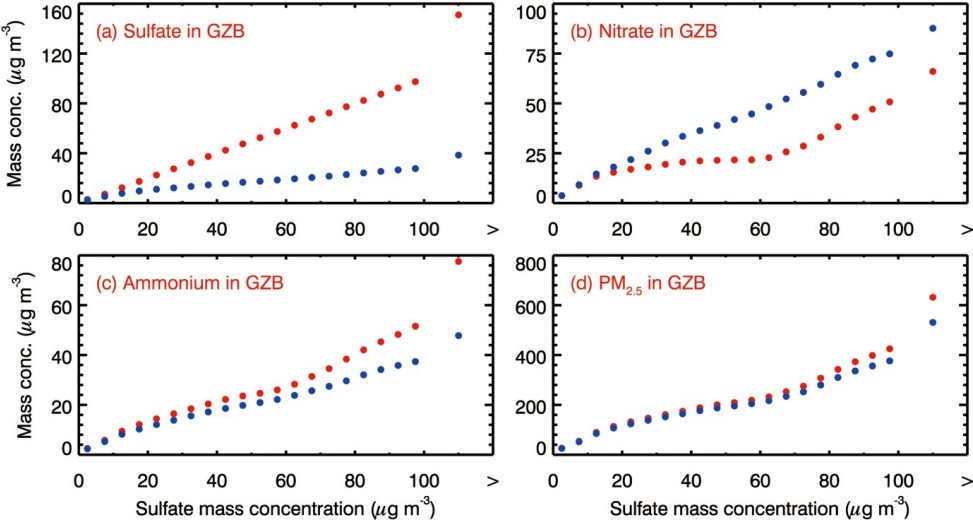

Figure 16 Average (a) sulfate, (b) nitrate, (c) ammonium, and (d) PM$_{2.5}$ mass concentrations
in GZB during the simulation period as a function of the sulfate mass concentration in the E-
case. The red and blue dots represent average mass concentrations in the E-case and B-case,
respectively.

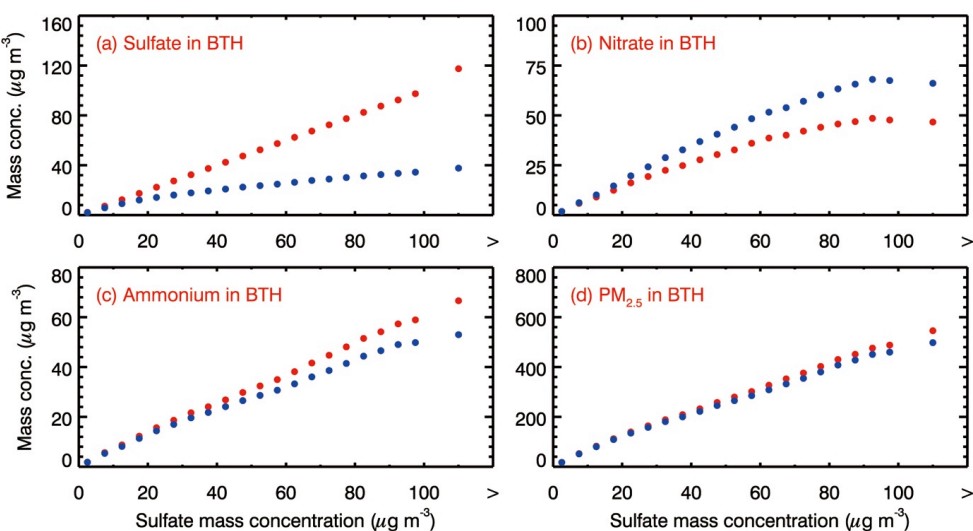

Figure 17 Same as Figure 16, but in BTH from 13 to 21 January 2014.