# Peer review of "A Possible Pathway for Rapid Growth of Sulfate during Haze Days in China"

_Atmospheric Chemistry and Physics, 2016_

## Referee Comment (RC1) · Anonymous Referee #1 · 22 Dec 2016

**General comments**

In this study, the authors attempted to implement a SO2 heterogeneous reaction parameterization into chemical transportation models to improve simulation of the sulfate rapid growth during haze pollution periods. The proposed parameterization focused on the treatment of the $Fe^{3+}$-catalyzed oxidation of SO2 by O2 in aerosol water. Simulations using WRF-CHEM model were conducted on haze episodes at two cities in China to evaluate the performance of the new parameterization. The authors found that the new parameterization could improve the representation of sulfate heterogeneous formation in WRF-CHEM model since the simulations with the parameterization could reproduce the observed rapid growth of sulfate aerosol and diurnal variations. Given that current models still underestimate the conversion of SO2 to sulfate, the SO2 heterogeneous reaction parameterization proposed here to improve sulfate simulation would be interesting to the readerships of the ACP journal. However, some issues related to the clarity of discussions and latest refs need to be addressed before its publication.

**Specific comments**

1) In the abstract and a statement on p. 3 lines 80-82, it appeared that the observed filter measurements in Xi'an, China since 2003 was used to develop SO2 heterogeneous reaction parameterization, but how to apply these filter measurements to parameterize SO2 heterogeneous reaction was not clearly explained. According to the parameterization section (section 3.1), it was more like that the filter measurements was only used to illustrate the relationships between sulfate, iron, humidity and PM2.5. None of values for the parameters in parameterization equation (line 208) was derived based on the filter measurements.

2) At several places (e.g., p. 7, lines 166-175; p. 9, lines 210-213) the authors stated that oxidation of sulfite by NO2 in aerosol water was proposed to contribute considerably to the sulfate production when NH3 concentrations were high. A very recent paper (Wang et al., PNAS, 2016, 113, 13630–13635) has provided the elucidation of this specific mechanism for sulfite-sulfate conversion. In addition, this work also pointed out the critical role of sulfate formation in haze development in China, including promoting the formation of SOA (Zhao et al., Environ. Sci. Technol. 40, 7682, 2006) and nitrate (Zhang et al., Geophys. Res. Lett. 22, 1493, 1995). Those references should be discussed when discussing the aerosol chemistry.

3) Some statements regarding to the discrepancies between simulation and observation were confused and speculative. For example, in both lines 293 and 314, it was stated that the model had difficulties in reproducing the long-range transport of pollutants like sulfate and nitrate. My concern was that the long-range transport contribution to pollutants could be negligible in this case since Guanzhong Basin was under the control of stagnation condition based on wind fields shown in Fig. 7. Therefore, the long-range transport may be not the reason why the simulated concentrations differ from the observations. Also in lines 302-307, the discrepancies between

simulated and observed sulfate mass was attributed to inaccurate simulations of wind fields, but there was no direct comparison between simulated and observed wind fields to demonstrate this point.

4) To be consistent with Figs. 11 and 12, how about adding one panel for time series of NH3 (in gas phase) to Figs. 9 and 10 to evaluate the model performance on NH3?

**Technical corrections**

1) On p. 5 in the equation for defining IOA, $|P_i - \bar{O}|$ in the denominator should be $|P_i - \bar{P}|$

2) Line 321: "sulfate aerosols play a more important role" than what? Nitrate aerosol?

---

## Referee Comment (RC2) · Anonymous Referee #2 · 1 Jan 2017

Review on "A Possible Pathway for Rapid Growth of Sulfate during Haze Days in China" by Li et al.

**General Comments**

This manuscript presents a heterogeneous formation mechanism of sulfate in which gaseous $SO_2$ is proposed to be oxidized by $O_2$ on the aerosol water surface with $Fe^{3+}$ serving as the catalyst. The parameterized mechanism is implemented in WRF-Chem and is evaluated through two heavy haze episodes in the China urban environments. Model simulations show that the proposed mechanism can reproduce the observed sulfate concentrations and improve the PM2.5 simulations. This study provides valuable information on improving our understanding of the $SO_2$ oxidation and sulfate formation in the atmosphere. It is well written and is suitable for publication with minor revisions.

**Specific comments**

1. There are two possible pathways for the heterogeneous $SO_2$ oxidation catalyzed by $Fe^{3+}$ involving aerosol water—aqueous reactions in cloud water or fog, and heterogeneous reactions on aerosol surfaces (e.g., lines 47-48 and lines 164-165). Does the proposed mechanism in this study consider both and only the latter? If it considers the latter only, would there still be some overlaps in the parameterization presented? And would the sulfate concentrations be possibly overestimated if both pathways are included? Are there any connections or relations between these two pathways?

2. L208, How is the aerosol water surface area calculated? Since the $SO_2$ oxidation is highly sensitive to RH, it is critical to treat the aerosol hygroscopic growth, which is closely relevant to the aerosol chemical composition, in the model. How is the aerosol hygroscopic growth treated in the model?

3. Lines 104-106, The two haze events need to be elaborated.

4. Lines 140-155, It would be helpful to provide quantitative contributions of the gas-phase oxidations by OH (and sCI if possible) to the sulfate formation.

5. Section 3.2, Given the evidence of the importance of RH in the $SO_2$ oxidation, it would be helpful to add the evaluation of the RH simulations and discussions of the effects of possible simulated RH biases.

6. Section 3.2, The authors attribute all modeled biases of sulfate concentrations to long range transport and/or meteorological factors. There may be other factors that also contribute to the biases (such as other oxidation mechanisms). Among the meteorology, RH could be a factor too.

7. Lines 244-350, It would be helpful to include percentage contributions of the HRSO2 mechanism for the two episodes.

**Technical comments**

1. Line 21, Should switch the order of develop and evaluate.

2. Line 66, "model oxides"?

3. Line 186, "showing considerable background contributions", of what, irons? PM2.5?

---

## Author Comment (AC1) · 21 Feb 2017

**Reply to Anonymous Referee #1**

We thank the reviewer for the careful reading of the manuscript and helpful comments. We have revised the manuscript following the suggestion, as described below.

**General comments**

In this study, the authors attempted to implement a $SO_2$ heterogeneous reaction parameterization into chemical transportation models to improve simulation of the sulfate rapid growth during haze pollution periods. The proposed parameterization focused on the treatment of the $Fe^{3+}$- catalyzed oxidation of $SO_2$ by $O_2$ in aerosol water. Simulations using WRF-CHEM model were conducted on haze episodes at two cities in China to evaluate the performance of the new parameterization. The authors found that the new parameterization could improve the representation of sulfate heterogeneous formation in WRF-CHEM model since the simulations with the parameterization could reproduce the observed rapid growth of sulfate aerosol and diurnal variations. Given that current models still underestimate the conversion of $SO_2$ to sulfate, the $SO_2$ heterogeneous reaction parameterization proposed here to improve sulfate simulation would be interesting to the readerships of the ACP journal. However, some issues related to the clarity of discussions and latest refs need to be addressed before its publication.

**Specific comments:**

**(1) Comment:** In the abstract and a statement on p. 3 lines 80-82, it appeared that the observed filter measurements in Xi'an, China since 2003 was used to develop $SO_2$ heterogeneous reaction parameterization, but how to apply these filter measurements to parameterize $SO_2$ heterogeneous reaction was not clearly explained. According to the parameterization section (section 3.1), it was more like that the filter measurements was only used to illustrate the relationships between sulfate, iron, humidity and $PM_{2.5}$. None of values for the parameters in parameterization equation (line 208) was derived based on the filter measurements.

**Response:** We have clarified in the abstract: "*The relationships based on the observed sulfate with PM$_{2.5}$, iron, and relative humidity in Xi'an, China have been employed to evaluate the mechanism and to develop a parameterization of the sulfate heterogeneous formation involving aerosol water for incorporation into atmospheric chemical transport models.*".

**(2) Comment:** At several places (e.g., p. 7, lines 166-175; p. 9, lines 210-213) the authors stated that oxidation of sulfite by NO$_2$ in aerosol water was proposed to contribute considerably to the sulfate production when NH$_3$ concentrations were high. A very recent paper (Wang et al., PNAS, 2016, 113, 13630–13635) has provided the elucidation of this specific mechanism for sulfite- sulfate conversion. In addition, this work also pointed out the critical role of sulfate formation in haze development in China, including promoting the formation of SOA (Zhao et al., Environ. Sci. Technol. 40, 7682, 2006) and nitrate (Zhang et al., Geophys. Res. Lett. 22, 1493, 1995). Those references should be discussed when discussing the aerosol chemistry.

**Response:** We have clarified in Section 3.1 as follows: "*Recently, Wang et al., (2016) have also elucidated a specific mechanism for the sulfite-sulfate conversion, in which oxidation of sulfite by NO$_2$ in aerosol water in case of high NH$_3$ concentrations contributes considerably to the sulfate production. They have also pointed out the critical role of the sulfate formation in haze formation in China through further promoting the formation of SOA and nitrate due to the enhanced hygroscopicity. Zhang et al. (1995) have reported that the high concentration of nitrate is attributed to an efficient heterogeneous conversion of NO$_x$ to HNO$_3$ due to the hydrolysis of N$_2$O$_5$ on sulfate aerosols. Zhao et al. (2006) have investigated the heterogeneous chemistry of methylglyoxal with liquid H$_2$SO$_4$, showing that the hydration and oligomerization reactions of methylglyoxal are enhanced by sulfate formation due to the high dependence of these reactions on particle hygroscopicity. Therefore, future studies need to be performed to incorporate the specific mechanism into CTMs to improve sulfate, nitrate, and SOA simulations.*"

Wang, G., Zhang, R., Gomez, M. E., Yang, L., Levy, Z. M., Hu, M., Lin, Y., Peng, J., Guo, S., and Meng, J.: Persistent sulfate formation from London Fog to Chinese haze,

Proceedings of the National Academy of Sciences of the United States of America, 113, 13630, 2016.

Zhang, R., Leu, M. T., and Keyser, L. F.: Hydrolysis of $N_2O_5$ and $ClONO_2$ on the $H_2SO_4/HNO_3/H_2O$ ternary solutions under stratospheric conditions, Geophysical Research Letters, 22, 1493-1496, 1995.

Zhao, J., Levitt, N. P., Zhang, R., and Chen, J.: Heterogeneous reactions of methylglyoxal in acidic media: implications for secondary organic aerosol formation, Environmental Science & Technology, 40, 7682-7687, 2006.

**(3) Comment:** Some statements regarding to the discrepancies between simulation and observation were confused and speculative. For example, in both lines 293 and 314, it was stated that the model had difficulties in reproducing the long-range transport of pollutants like sulfate and nitrate. My concern was that the long-range transport contribution to pollutants could be negligible in this case since Guanzhong Basin was under the control of stagnation condition based on wind fields shown in Fig. 7. Therefore, the long-range transport may be not the reason why the simulated concentrations differ from the observations. Also in lines 302-307, the discrepancies between simulated and observed sulfate mass was attributed to inaccurate simulations of wind fields, but there was no direct comparison between simulated and observed wind fields to demonstrate this point.

**Response:** We have removed the speculative sentences in both lines 293 and 314, and lines 302-307 due to lack of comparisons of simulated wind fields with observations. We have also included a paragraph about the evaluation of RH in Xi'an and Beijing and the impact of the simulated RH uncertainties on the sulfate simulations:

" *Considering the importance of RH in the $SO_2$ heterogeneous oxidation, Figure 13 shows the simulated and observed RH in Xi'an from December 16 to 27, 2013 and in Beijing from January 13 to 21, 2014. The model generally performs reasonably well in simulating the observed RH, with IOAs of 0.80 for Xi'an and 0.76 for Beijing. Overall, the model is subject to overestimate the RH, especially in Beijing, but well captures the observed peaks of the RH in Beijing and Xi'an. The RH biases considerably affect the sulfate simulations. The underestimation of the high RH generally corresponds the underestimation of the sulfate*

*concentration, i.e., during nighttime on January 15 and 16, 2014 in Beijing, and in the morning from December 23 to 25, 2013 in Xian.".*

**(4) Comment:** To be consistent with Figs. 11 and 12, how about adding one panel for time series of NH$_3$ (in gas phase) to Figs. 9 and 10 to evaluate the model performance on NH$_3$?

**Response:** We do not have the NH$_3$ measurement from 16 to 27 December 2013 in GZB and from 13 to 21 January 2014 in BTH. We have clarified in Section 3.2: "*Due to lack of routine measurements of NH$_3$ in GZB and BTH, the evaluation of the model performance on NH$_3$ is not provided in the present study. Future studies are imperative to be performed to evaluate the model performance on NH$_3$ which plays an important role in the sulfate formation (Wang et al., 2017).*"

**Technical corrections**

**Comment:** On p. 5 in the equation for defining IOA, $|P_i\text{-}\overline{O}|$ in the denominator should be $|P_i - \overline{P}|$

**Response:** The equation of IOA in the manuscript is right, please reference the website "http://www.rforge.net/doc/packages/hydroGOF/d.html".

**Comment:** Line 321: "sulfate aerosols play a more important role" than what? Nitrate aerosol?

**Response:** We have revised the sentence as "*sulfate aerosols play an important role*" in Section 3.2.

---

## Author Comment (AC2) · 21 Feb 2017

**Reply to Anonymous Referee #2**

We thank the reviewer for the careful reading of the manuscript and helpful comments. We have revised the manuscript following the suggestion, as described below.

**General Comments**

This manuscript presents a heterogeneous formation mechanism of sulfate in which gaseous $SO_2$ is proposed to be oxidized by $O_2$ on the aerosol water surface with $Fe^{3+}$ serving as the catalyst. The parameterized mechanism is implemented in WRF-Chem and is evaluated through two heavy haze episodes in the China urban environments. Model simulations show that the proposed mechanism can reproduce the observed sulfate concentrations and improve the $PM_{2.5}$ simulations. This study provides valuable information on improving our understanding of the $SO_2$ oxidation and sulfate formation in the atmosphere. It is well written and is suitable for publication with minor revisions.

**Specific comments**

**(1) Comment:** There are two possible pathways for the heterogeneous $SO_2$ oxidation catalyzed by $Fe^{3+}$ involving aerosol water—aqueous reactions in cloud water or fog, and heterogeneous reactions on aerosol surfaces (e.g., lines 47-48 and lines 164-165). Does the proposed mechanism in this study consider both and only the latter? If it considers the latter only, would there still be some overlaps in the parameterization presented? And would the sulfate concentrations be possibly overestimated if both pathways are included? Are there any connections or relations between these two pathways?

**Response:** We have highlighted in Section 3.1: "*There are two possible pathways for the sulfate formation: bulk aqueous-phase oxidation of $SO_2$ in aerosol water and heterogeneous reaction of $SO_2$ on aerosol surfaces involving aerosol water.*", and further clarified in Section 3.1: "*We propose here a $SO_2$ heterogeneous reaction parameterization in which the $SO_2$ oxidation in aerosol water by $O_2$ catalyzed by $Fe^{3+}$ is limited by mass transfer resistances in the gas-phase and the gas-particle interface.*". So, The proposed mechanism considers both the aqueous-phase oxidation of $SO_2$ in aerosol water and heterogeneous reaction of $SO_2$ on aerosol surfaces involving aerosol water.

The $Fe^{3+}$ catalytic reaction occurring in the two $SO_2$ oxidation pathways is the same, but under different circumstances. The sulfate concentrations are not possibly overestimated when the both pathways are included.

**(2) Comment:** Line-208, How is the aerosol water surface area calculated? Since the $SO_2$ oxidation is highly sensitive to RH, it is critical to treat the aerosol hygroscopic growth, which is closely relevant to the aerosol chemical composition, in the model. How is the aerosol hygroscopic growth treated in the model?

**Response:** We have clarified in Section 3.1: "*The aerosol hygroscopic growth is directly predicted by ISORROPIA (Version 1.7) in the model and the aerosol water surface area is scaled from the calculated wet aerosol surface area using the third-moment of aerosol species.*". Considering that the $SO_2$ heterogeneous oxidation is highly sensitive to RH, we have further evaluated evaluations of the RH simulations (Please found in Comment 5).

**(3) Comment:** Lines 104-106, The two haze events need to be elaborated.

**Response:** We have added the description of the two haze events in Section 2.1 as follows: "*A very severe haze episode occurred in GZB during the period from December 16 to 27, 2013, with an average $PM_{2.5}$ concentration of 325.6 µg $m^{-3}$. The maximum of the average $PM_{2.5}$ concentration in GZB even exceeded 500 µg $m^{-3}$ during the episode. The average temperature and relative humidity in Xi'an was 3.7 ℃ and 72% during the episode, respectively, and the average wind speed was around 3.7 m $s^{-1}$. The average $PM_{2.5}$ concentration from January 13 to 21, 2014 in BTH was 195.3 µg $m^{-3}$, with a maximum of 363.9 µg $m^{-3}$. The average temperature and relative humidity in Beijing during the episode was -0.5 ℃ and 42%, respectively, and the average wind speed was about 7.4 m $s^{-1}$.*"

**(4) Comment:** Lines 140-155, It would be helpful to provide quantitative contributions of the gas-phase oxidations by OH (and sCI if possible) to the sulfate formation.

**Response:** We have included the discussion in Section 3.2: "*It is worthy to note that during the two episodes, the SO$_2$ oxidation by OH to the sulfate formation is not important. We have performed additional sensitivity simulations in which only the direct emissions of sulfate are considered. Comparisons of the sensitivity simulation with the B-case show that the SO$_2$ oxidation by OH can explain about 5.1% and 11.7% of the observed sulfate concentrations in Xi'an and Beijing on average, respectively.*"

**(5) Comment:** Section 3.2, Given the evidence of the importance of RH in the SO$_2$ oxidation, it would be helpful to add the evaluation of the RH simulations and discussions of the effects of possible simulated RH biases.

**Response:** We have added Figure 13 in the Section 3.2 and clarified as follows:

" *Considering the importance of RH in the SO$_2$ heterogeneous oxidation, Figure 13 shows the simulated and observed RH in Xi'an from December 16 to 27, 2013 and in Beijing from January 13 to 21, 2014. The model generally performs reasonably well in simulating the observed RH, with IOAs of 0.80 for Xi'an and 0.76 for Beijing. Overall, the model is subject to overestimate the RH, especially in Beijing, but well captures the observed peaks of the RH in Beijing and Xi'an. The RH biases considerably affect the sulfate simulations. The underestimation of the high RH generally corresponds the underestimation of the sulfate concentration, i.e., during nighttime on January 15 and 16, 2014 in Beijing, and in the morning from December 23 to 25, 2013 in Xian.*"

**(6) Comment:** Section 3.2, The authors attribute all modeled biases of sulfate concentrations to long range transport and/or meteorological factors. There may be other factors that also contribute to the biases (such as other oxidation mechanisms). Among the meteorology, RH could be a factor too.

**Response:** We have removed the speculative sentences in Section 3.2 due to lack of comparisons of simulated wind fields with observations and included the discussion about the effect of the simulated RH biases on the sulfate simulations.

**(7) Comment:** Lines 244-350, It would be helpful to include percentage contributions of the HRSO$_2$ mechanism for the two episodes.

**Response:** We have classified in Section 3.2 as follows:

*"The difference of the simulated [NO$_2$] in the B-case and E-case is minor, and the average [NO$_2$] is increased by 0.69% in GZB and decreased by 0.1% in BTH in the E-case compared to the B-case, showing that the impact of the HRSO$_2$ parameterization on NO$_2$ simulations is not significant in GZB and BTH."*

*"On average, inclusion of the HRSO$_2$ parameterization decreases the [SO$_2$] by 15.9% and 3.4% in GZB and BTH on average, respectively."*

*"However, in the E-case, the WRF-CHEM model generally yields the observed sulfate variations during the 11-day episode, with a MB of -17.0 µg m$^{-3}$ and an IOA of 0.89, and the average sulfate concentration is enhanced by 172% compared to the B-case."*

*"The average sulfate concentration is enhanced by 58.4% in the E-case compared to the B-case in Beijing."*

*"The inclusion of the HRSO$_2$ parameterization deceases the simulated nitrate concentration by 15.3% and 19.5% in Xi'an and Beijing, respectively, on average."*

*"The average ammonium concentration is enhanced by 36.8% in Xi'an and 7.2% in Beijing by the inclusion of the HRSO$_2$ parameterization."*

*"Inclusion of the HRSO$_2$ parameterization in the E-case improves the ability of the model to reproduce the PM$_{2.5}$ measurements in GZB and BTH. In GZB, due to very humid conditions which facilitate the heterogeneous sulfate formation during the episode, the simulated PM$_{2.5}$ mass concentrations are increased by more than 40 µg m$^{-3}$ in the E-case compared to the B-case with an average increase of 12.3%, and more consistent with the measurements. The HRSO$_2$ parameterization also improves the PM$_{2.5}$ simulations in BTH,*

*with an average increase of less than 3.0%, reducing the underestimation from around*
*-13.3 to -5.1 μg m$^{-3}$."*

**Technical comments**

**Comment:** Line 21, Should switch the order of develop and evaluate.

**Response:** The relationships obtained from observed sulfate with PM2.5, iron, and relative humidity are first used to evaluate the proposed mechanism and further, based on the mechanism to develop a sulfate heterogeneous parameterization.

**Comment:** Line 66, "model oxides"?

**Response:** We have changed the "model oxides" as "*oxides*" in the manuscript.

**Comment:** Line 186, "showing considerable background contributions", of what, irons? PM$_{2.5}$?

**Response:** We have revised the sentence in Section 3.1 as "*showing considerable background iron contributions*".